# Comparative efficacy and acceptability of psychotherapies for post-traumatic stress disorder in children and adolescents: study protocol for a systematic review and network meta-analysis

Yuqing Zhang,[1,2] Xinyu Zhou,[3] Lining Yang,[1,2] Sarah E Hetrick,[4,5] John R Weisz,[6] Pim Cuijpers,[7] Jürgen Barth,[8] Cinzia Del Giovane,[9] Shuai Yuan,[1,2] David Cohen,[10] Donna Gillies,[11] Xiaofeng Jiang,[1,2] Teng Teng,[1,2] Peng Xie[1,2]

YZ, XZ and LY contributed equally.

For numbered affiliations see end of article.

**Correspondence to**
Professor Peng Xie;
xiepeng973@126.com

## ABSTRACT

**Introduction** Post-traumatic stress disorder (PTSD) is common among children and adolescents who are exposed to trauma, and it is often associated with significant negative impacts on their psychosocial functioning and quality of life. Many types of psychotherapies have been found to be effective for PTSD in children and adolescents. However, due to the lack of direct comparisons between different psychotherapies, the hierarchy of treatment efficacy is still unclear. Therefore, we plan to conduct a systematic review and network meta-analysis to evaluate the efficacy and acceptability of various types of psychotherapies for PTSD in children and adolescents.

**Methods and analysis** A systematic search will be conducted among eight electronic databases, including PubMed, Cochrane, Embase, Web of Science, PsycINFO, Cumulative Index of Nursing and Allied Health, Published International Literature on Traumatic Stress (PILOTS) and ProQuest Dissertations, from inception to October 2017. Randomised controlled trials, regardless of language, publication year and publication type, comparing any psychotherapies for PTSD to any control condition or alternative treatment in children and adolescents (18 years old or less) diagnosed with full or subclinical PTSD will be included. Study duration and the number of treatment sessions will not be limited. The primary outcome will be PTSD symptom severity at post-treatment as measured by a rating scale reported by the child, parent or a clinician. The secondary outcomes will include: (1) efficacy at follow-up; (2) acceptability (all-cause discontinuation); (3) anxiety symptom severity; (4) depressive symptom severity and (5) quality of life and functional improvement. Bayesian network meta-analyses for all relative outcome measures will be performed. We will conduct subgroup and sensitivity network meta-analyses to determine whether the findings are affected by study characteristics. The quality of the evidence contributing to network estimates of the primary outcome will be evaluated by the Grading of Recommendations, Assessment, Development and Evaluations framework.

### Strengths and limitations of this study

► Bayesian network meta-analysis can simultaneously compare various types of treatments by integrating all the best evidence (direct and indirect evidence) to estimate the interrelations across all treatments and establish a treatment hierarchy for psychotherapies for post-traumatic stress disorder (PTSD) in children and adolescents.

► A number of outcomes will be used to comprehensively assess efficacy at post-treatment and follow-up, acceptability, anxiety and depressive symptom severity, quality of life and functional improvement.

► This study will help to guide clinical decision-making regarding the relative efficacy of various types of psychotherapy and various delivery modalities in the treatment of PTSD in children and adolescents.

► Subgroup and sensitivity network meta-analyses will help to find potential moderators that affect the efficacy of psychotherapies.

► The limitations of included studies will be assessed using the Cochrane Collaboration's risk of bias V.2.0 tool, and the quality of evidence for network estimates of the primary outcome will be evaluated by the Grading of Recommendations, Assessment, Development and Evaluations framework.

**Ethics and dissemination** No ethical issues are foreseen. The results will be published in a peer-reviewed journal, which will be disseminated electronically and in print. This network meta-analysis may be updated to inform and guide the clinical management of PTSD in children and adolescents.

**PROSPERO registration number** CRD42016051786.

## BACKGROUND

Many children and adolescents are exposed to trauma, with more than two-thirds

reporting at least one traumatic event by 16 years of age.[1] Post-traumatic stress disorder (PTSD) is one of the most common mental disorders among children and adolescents who have experienced trauma. Data from a meta-analysis pooling 72 articles with 3563 trauma-exposed children and adolescents show that the overall rate of PTSD was 15.9%.[2] As described in the Diagnostic and Statistical Manual of Mental Disorders, Fifth edition (DSM-5), PTSD diagnostic criteria includes four symptom clusters: (1) re-experiencing of the traumatic event; (2) avoidance of stimuli associated with the traumatic event; (3) negative alterations in cognition and mood and (4) hyperarousal.[3] Trauma exposure is an essential factor to be able to diagnose PTSD, which can include physical or sexual abuse, war or terrorism, natural or man-made disasters, witnessing domestic violence, catastrophic illnesses, vehicle or other accidents.[4 5] PTSD is correlated with various adverse consequences for children and adolescents in cognitive, emotional, social, academic and other functional domains.[6] In addition, PTSD commonly co-occurs with other psychiatric conditions, such as major depressive disorder,[7] anxiety disorder,[7] substance abuse[8] and attention deficit hyperactivity disorder (ADHD).[9] PTSD diagnosis is accompanied by a significant reduction in quality of life, and it is estimated that successful treatment could save 2.05 quality adjusted life years per child or adolescent with PTSD.[10]

Several forms of interventions have been applied to address PTSD symptoms in children and adolescents, from which psychotherapy is the mainstay of treatment.[11] Although some medications (eg, fluoxetine, paroxetine and venlafaxine) have shown a significant but small superiority over placebo (mean effect size=0.23) in PTSD treatment for adults,[12] their use in children and adolescents is still limited because (1) the evidence of the effectiveness of pharmacological treatments for children and adolescents with PTSD is scant[13] and (2) medications have been associated with safety concerns in children and adolescents, including evidence of increased risk of suicidality in children and adolescents treated with antidepressants.[14 15] Therefore, clinical guidelines recommend that psychotherapy should be used for the initial treatment of PTSD in children and adolescents.[16]

Since the 1990s, psychotherapies have been increasingly applied in the treatment of PTSD in children and adolescents, and recent reviews have reported encouraging results on their efficacy.[17–19] Evidence from Cochrane reviews found that psychological therapies are effective in preventing PTSD in children and adolescents who have undergone trauma and effective in treating PTSD in children and adolescents.[17 18] Many types of psychotherapies are now available, such as cognitive behavioural therapy (CBT), eye movement desensitisation and reprocessing (EMDR), behavioural therapy (BT), cognitive therapy, psychodynamic therapy and play therapy. Among these, trauma-focused CBT (TF-CBT), a CBT programme that uses cognitive-behavioural techniques with a trauma-focused component, is the most commonly practised

psychotherapy for children and adolescents with PTSD.[20] TF-CBT has been recommended as a first-line treatment by clinical guidelines for PTSD in children and adolescents.[16 21] However, trauma-focused therapy can be difficult for some therapists to implement,[22 23] and there are some safety concerns in clinical practice (eg, symptoms worsening and patients ending treatment prematurely).[24 25] Two recent trials that directly compared TF-CBT with non-TF-CBT alternatives reported that both were efficacious.[23 26] Thus, whether the inclusion of a trauma-focused component is essential in CBT for PTSD in children and adolescents is still uncertain. Moreover, other therapies, such as EMDR and BT, have also been reported effective in treating PTSD symptoms in children and adolescents.[27 28] However, due to methodological limitations, conventional meta-analyses cannot simultaneously compare all these treatments. Therefore, they cannot provide a clear answer regarding the best choice for initial treatment nor can they provide a hierarchy of these psychotherapies.

Network meta-analysis is a newly developed method for evidence synthesis, which is able to integrate direct evidence (from studies directly comparing interventions) with indirect evidence (derived from separate studies addressing a common reference condition) from multiple treatment comparisons to estimate the interrelations across all treatments.[29] This approach enables a simultaneous comparison of multiple treatments and can provide hierarchical evidence to guide clinical practice. Using this method, our group has produced meta-analyses comparing the effects of different psychotherapies for the treatment of anxiety disorder[30] and depressive disorder[31] in children and adolescents. In the present study, we describe the methods to undertake a network meta-analysis to complement those previous reports that will focus on the efficacy of psychotherapies for children and adolescents with PTSD. By performing a well-designed Bayesian network meta-analysis, we aim to provide a higher level of evidence that can inform clinical guidelines for PTSD treatment in children and adolescents.

## METHODS
### Criteria for included studies
#### Types of studies
All randomised controlled trials (RCTs), including cluster-randomised trials and cross-over trials, will be included. However, only the results from the first randomisation period will be considered in the cross-over trials. For the purpose of reducing heterogeneity between trials, we will exclude quasirandomised trials (eg, allocation based on the last number of the date of birth) and trials in which the sample size is less than 10 per study. Study duration and the number of treatment sessions will not be limited.

## Types of participants

Studies that enrolled children and adolescents, aged 18 years old or less when they were initially enrolled, will be included in this review. Given that children with significant PTSD symptoms who do not meet full criteria for a PTSD diagnosis often have comparable functional impairment to those with a PTSD diagnosis,[32–34] the clinical guideline suggested that treatment decisions should be based on symptom severity and functional impairment rather than whether or not they have an actual PTSD diagnosis.[16] Therefore, we will apply the following broad criteria to identify the participants: (1) Full PTSD, as diagnosed according to standardised diagnostic interviews based on international classifications (DSM,[3 35–38] the International Classification of Diseases (ICD)[39 40] or validated scales for PTSD based on DSM/ICD criteria)[41–44]; (2) subclinical/partial PTSD, defined as patients who have experienced psychological trauma and report some subsequent PTSD symptoms in at least one of the four symptom clusters according to DSM-5 (ie, re-experiencing, avoidance, hyperarousal, negative alterations in cognition and mood)[45]; (3) clinically significant post-traumatic stress symptoms, defined as scoring above a validated cut-off on a PTSD rating scale, such as the Kiddie Schedule for Affective Disorders and Schizophrenia,[42] Child PTSD Symptoms Scale[46] and the Impact of Event Scale.[47] We will include trials in which participants have a secondary diagnosis of comorbid general psychiatric disorders, for example, depressive disorder, anxiety disorder, ADHD and oppositional defiant disorder. However, trials in which participants have a diagnosis of acute stress disorder or adjustment disorder will be excluded. Studies where both adults and children/adolescents are included will be eligible for inclusion if the data for the latter can be obtained separately. Studies where it is not clear what happened to the patients who withdrew from the study will not be excluded, and all these patients will be counted as all-cause discontinuation. Trials conducted in any treatment settings, including outpatient clinics, inpatient services, community clinics and schools will be included.

## Types of interventions

All RCTs comparing any psychotherapy against another psychotherapy or any control condition for children and adolescents with PTSD will be included. We will view each of: TF-CBT, non-trauma-focused CBT (non-TF-CBT), EMDR, BT, cognitive therapy, psychodynamic therapy, play therapy, and any other psychotherapy as independent nodes in this network meta-analysis. Trials comparing the same type of psychotherapies, but at different delivery conditions (with or without family involvement), different delivery formats (group, individual or group plus individual) and different delivery mediums (face-to-face, internet-based) will be considered as the same node in this network meta-analysis. Nevertheless, because TF-CBT is the most commonly studied psychotherapy and recommended by clinical guidelines as first-line choice for children and adolescents with PTSD,[16 21] if data available, we will separate TF-CBT with different delivery conditions, formats and mediums as independent nodes.

Control conditions can include waitlist, non-treatment and treatment as usual. We will view each of these control conditions as independent nodes in this network meta-analysis. The detailed description of each treatment and control condition are presented in table 1. According to the principles described in this table, two reviewers will independently perform the classification of all conditions in each trial. Any disagreements will be discussed among the review team, and any unclear information will be requested from the relevant authors.

Studies where psychotherapy is used as a combination strategy (eg, combining psychotherapy with medication) will be excluded, because such designs make it impossible for us to estimate the effect size of each specific treatment approach. We will not exclude studies that enrolled patients who had used medications in the past, provided that their medication status was not changed for at least 1 month prior to study entry and for the study period.

## Types of outcome measures
### Primary outcome

1. Efficacy at post-treatment, as measured using the endpoint score from PTSD symptom severity rating scales completed by the child, parent or a clinician.[48] When end-point scores are not reported, we will use change scores (if available).[49] Where more than one scale is reported in a trial, we will extract data from the PTSD symptom scales on a hierarchy, which is based on psychometric properties and appropriateness for use with children and adolescents (table 2). In addition, where a PTSD symptom scale is reported by different raters in a trial, self-rated outcome will be preferred, then the parent or clinician rated outcome, because self-rated outcome tends to result in more conservative effect sizes.[50] Where dichotomous efficacy outcomes, instead of continuous scores, are reported in a trial, we will contact the relevant authors to request the data we need. If they did not respond the data will not be used.

### Secondary outcomes

1. Efficacy at follow-up, as measured by the score from PTSD scales at the longest point of follow-up (up to 12 months). The selection priority of PTSD scales will be the same as for efficacy at post-treatment. Data from participants who take part in subsequent treatments (eg, continuing psychotherapy/pharmacotherapy or booster sessions) will be excluded in the follow-up analysis.
2. Acceptability, defined as all-cause discontinuation, as measured by the proportion of patients who discontinued treatment for any cause at the end of treatment. Notably, children and adolescents may discontinue treatment for many different reasons, such as finding it difficult to adhere to long-term treatment,

**Table 1** Description of psychotherapeutic interventions and control condition

| Interventions | Abbreviation | Description |
|---|---|---|
| **Psychotherapeutic intervention** | | |
| Trauma-focused cognitive behavioural therapy | TF-CBT | CBT is a combination of cognitive and behavioural techniques. It also involves additional techniques such as relaxation training, affective modulation skills and enhancement of future safety and development. TF-CBT is a CBT programme that involves a trauma focus, which is usually performed through exposure or cognitive processing of thoughts related to the trauma. |
| Non-trauma-focused cognitive behavioural therapy | Non-TF-CBT | Non-TF-CBT is a CBT programme that focuses on teaching skills for the reduction of anxiety. These treatments use procedures that directly target the person's beliefs and behaviours rather than the discussions of specific traumas. |
| Cognitive therapy | CT | CT mainly uses cognitive restructuring training, which aims at examining youths' automatic thoughts and core schemas and evaluating the accuracy and affective consequences of their views. They aim to teach youths to engage in 'rational' thinking about themselves, the traumatic incident and the world. |
| Behavioural therapy | BT | BT uses some form of behavioural training, especially for exposure -based therapy and narrative therapy, to help youth reduce trauma-related symptoms. BT is based on principles of habituation. |
| Eye movement desensitisation and reprocessing | EMDR | EMDR aims to help a person reprocess their memories of a traumatic event. The therapy involves bringing distressing trauma-related images, beliefs and bodily sensations to mind. |
| Psychodynamic therapy | DYN | Psychodynamic psychotherapy focuses on integrating the traumatic experience into the life experience of the individual as a whole. Childhood issues are often felt to be important. |
| Play therapy | PT | PT used techniques to engage participants in recreational activities to help them cope with their problems and fears. |
| Stress management | SM | SM mainly includes some form of relaxation or biofeedback |
| Supportive therapy | ST | ST is an unstructured therapy without specific psychological techniques that it helped people to ventilate their experiences and emotions and offering empathy, for example, supportive counselling, attention control, minimal contact, active listening, common factor control, non-specific control. |
| **Control conditions** | | |
| Treatment as usual | TAU | TAU is often described as 'usual care' or 'usual community treatment' in trials, which may include any components of psychotherapy or pharmacotherapy for PTSD. It is not considered to be structured intervention but may have some treatment effects. |
| Waitlist | WL | WL is a control condition in which the participants receive no active treatment during the study but are informed that they can receive one after the study period is over. |
| No treatment | NT | NT is a control condition in which the participants receive no active treatment during the study and in which they do not expect to receive such after the study is over. |

because symptoms worsen, or due to rapid improvement of symptoms.[30]

3. Anxiety symptoms, as measured by the end-point score on anxiety symptom severity rating scales. The following scales will be used: Revised Children's Manifest Anxiety Scale, Spence Children's Anxiety Scale, Multidimensional Anxiety Scale for Children, State-Trait Anxiety Inventory for Children, Screen for Anxiety and Related Disorders. If none of above scales are reported, other valid anxiety scales will be used. When a scale is reported by different raters in a trial, self-rated outcome will be preferred.

4. Depressive symptoms, as measured by the end-point score on depressive symptom severity rating scales. The following scales will used: Children's Depression Inventory, Beck depression inventory, Mood and Feeling Questionnaire, Children's Depression Rating Scale Revised, Hamilton Depression Rating Scale. As for anxiety, other valid depression scales will be used if none of the above scales were reported.

5. Quality of life and functional improvement (QoL/functioning). If data are available, we will extract continuous outcomes from scales of QoLand functional improvement measured at post-treatment. The

**Table 2** Hierarchy of PTSD symptom severity measurement scales

| Hierarchy | PTSD symptom severity rating scales | Abbreviation |
|---|---|---|
| 1 | UCLA Post-Traumatic Stress Disorder Reaction Index | UCLA PTSD Index |
| 2 | Child PTSD Symptoms Scale | CPSS |
| 3 | Clinician-Administered PTSD Scale/Clinician Administered PTSD Scale-Child and Adolescent Version | CAPS/CAPS-CA |
| 4 | Impact of Events Scale/The Children's Revised Impact of Events Scale | IES/CRIES |
| 5 | Parent Report of Post-traumatic Symptoms/Child Report of Post-traumatic Symptoms | PROPS/CROPS |
| 6 | Kiddie-Schedule for Affective Disorders and Schizophrenia | K-SADS |
| 7 | Trauma Symptom Checklist for Children | TSCC |
| 8 | Post-Traumatic Cognitions Inventory/Child Post-traumatic Cognitions Inventory | PTCI/CPTCI |
| 9 | Harvard Trauma Questionnaire | HTQ |
| 10 | Post-traumatic Stress Scale | PSS |
| 11 | Child Post-Traumatic Stress—Reaction Index | CPTS-RI |
| 12 | The Preschool Age Psychiatric Assessment | PAPA |
| 13 | Anxiety Disorders Interview Schedule | ADIS |

UCLA, University of California, Los Angeles.

scales of QoL include Quality of Life Enjoyment and Satisfaction Questionnaire, Paediatric Quality of Life Inventory (QoL Child Report), the Quality of Life Inventory and others. The scales of functional improvement include the Children's Global Assessment Scale, the Global Assessment Functioning and others. When scales of quality of life and functional improvement are both reported in a trial, we will extract the data of quality of life.

### Search strategy

We will identify relevant trials from systematic searches in the following electronic databases: PubMed, Cochrane, Embase, Web of Science, PsycINFO, Cumulative Index of Nursing and Allied Health, PILOTS and ProQuest Dissertations. The timeline will be from database inception to October 2017. An example of search strategy for PubMed is included in the online supplementary material. In addition, international trials registers, such as WHO's trials portal, ClinicalTrials.gov and Australian New Zealand Clinical Trials Registry will be searched for ongoing trials. Furthermore, we will search the presentations at European Psychiatric Association, European College of Neuropsychopharmacology, American Psychological Association, American Psychiatric Association and European Society for Child and Adolescent Psychiatry congresses and hand search relevant key psychiatric, psychological and medical journals (including *Behavior Therapy, Child Abuse and Neglect, Child Maltreatment, Journal of Counseling and Development, Counseling Outcome Research and Evaluation, Journal of Anxiety Disorders, Journal of Mental Health Counseling, Journal of Trauma Practice, Journal of Traumatic Stress, Journal of the American Academy of Child and Adolescent Psychiatry, Journal of Consulting and Clinical Psychology, Behaviour research and therapy*). The reference lists of included trials and reviews identified from initial searches will be scanned for more relevant studies. All relevant authors will be contacted to complement the incomplete data.

### Study selection and data extraction
#### Selection of trials

Two reviewers (YZ and LY) will independently identify potentially eligible studies from the titles and abstracts of records from the search strategies. Studies will be excluded if both reviewers consider that it does not meet eligibility criteria. Then, the full texts of these potentially eligible studies will be reviewed by the same criteria. The inter-rater reliability of the two reviewers will be calculated to detect their consistency. All disagreements will be discussed and resolved by a senior review author (PX or SEH). The references of relevant reviews and included trials will be checked by YZ and LY. Where multiple publications derive from a common dataset, we will select the trials in which the relevant outcomes we predefined in this protocol were reported completely. We will report the reasons for exclusion for each trial in the characteristics of excluded studies list. Finally, a flow chart will be used to present the process of trial screening in this meta-analysis.

### Risk of bias assessment

Two reviewers (YZ and XZ) will independently assess the methodological quality of the included studies. According to the Cochrane Collaboration's risk of bias V.2.0 tool,[51] the risk of bias will be rated as 'low risk', 'high risk' or 'some concerns' in the following domains: (1) bias arising from the randomisation process; (2) bias due to deviations from intended interventions; (3) bias due to missing outcome data; (4) bias in measurement of the outcome; (5) bias in selection of the reported result and (6) overall bias. The inter-rater reliability of the two reviewers assessing the risk of bias will also be calculated.

Any disagreements will be resolved by a senior review author (PX or SEH).

## Data extraction

A standardised data extraction form will be used by two independent reviewers (YZ and XZ) to record the relevant parameters from the original paper, including study characteristics (eg, title, first author, publication year, publication type, publication journal, location and sponsor), patient characteristics (eg, diagnostic criteria for PTSD, type of trauma, severity of PTSD symptoms, comorbidities and the number, mean age and gender of participants), intervention details (eg, type of psychotherapy, treatment format, treatment setting, treatment duration, the number of sessions, follow-up duration and cointerventions) and outcome measures (mean scores, number of participants and outcome raters (ie, self-rated or observer rated) for each predefined outcome). A table will be used to present the main characteristics of each study included in this review. The reliability of data extraction from the reviewers will also be assessed. Any disagreements will be resolved by a senior review author (PX or SEH).

## Statistical analysis

We will perform Bayesian network meta-analysis for each outcome with random-effects model in WinBUGS V.1.4.3. In addition, as a reference for relative outcomes of network meta-analyses, conventional pairwise meta-analyses will also be performed for each outcome with a random-effects model in Stata V.13.0. Standardised mean difference (SMD, Cohen's d) will be used as a measure of effect size where outcome is measured on different scales. For dichotomous outcome (all-cause discontinuation), the effect sizes will be calculated as ORs. Statistical heterogeneity in each pairwise comparison will be assessed with the $I^2$ statistic and p value.[52] When the mean values or SDs of continuous outcomes are missing, we will compute values by conversion from p values, t-values, CIs or SEs.[53] Missing continuous outcome data will be analysed using completer data.

The pooled estimates of network meta-analysis will be obtained using the Markov Chains Monte Carlo method. Two Markov chains will be run simultaneously with different arbitrarily chosen initial values. Trace plots and the Brooks-Gelman-Rubin statistic will be monitored to ensure convergence.[54] When the model is adequately convergent, the foregoing samples will be discarded. Then 100 000 subsequent simulations will be run as the posterior summaries. All results will be reported as effect sizes (SMD or OR) and their 95% credible intervals (CrI). We will assume a common heterogeneity parameter for each comparison and assess the global heterogeneity using the $I^2$ statistic with 'gemtc' package in R V.3.2.2. Furthermore, the inconsistency between direct and indirect evidence will be evaluated, which will include the assessment of global inconsistency (by comparing the fit and parsimony of consistency and inconsistency models), local inconsistency (by calculating the difference among direct and indirect evidence in closed loops in the network)[55] and the inconsistency calculated by

node splitting method (by assessing the difference between direct and indirect estimates within a particular comparison).[56] Probability values will be summarised and reported as surface under the cumulative ranking curve to provide a hierarchy of the treatments.[57]

## Subgroup analysis

Where possible, we will conduct network meta-regression of data on primary outcome to evaluate the influence of the following potential moderators: (1) age group; (2) sex ratio; (3) number of sessions; (4) sample size; (5) risk of bias; (6) trauma types (acute/single trauma vs chronic trauma); (7) diagnosis criteria (youth with a diagnosis vs with subsyndromal symptoms); (8) source of outcome measure (self-rated vs observer rated). If the data are insufficient to conduct subgroup analyses for some moderators, we will perform sensitivity analyses by omitting specific trials from the overall analysis.

## Other analyses

Comparison-adjusted funnel plots and Egger's test will be used for each outcome to examine whether there is dominant publication bias exists in this network meta-analysis.[56] In addition, we will evaluate the quality of evidence for primary outcome by using the Grading of Recommendations, Assessment, Development and Evaluations framework, which characterises the quality of a body of evidence on the basis of the study limitations, imprecision, inconsistency, indirectness and publication bias for network estimates.[58]

## DISCUSSION

This systematic review and network meta-analysis will comprehensively evaluate the comparative efficacy and acceptability of psychological interventions for children and adolescents with PTSD. The results of this research will provide evidence to inform a hierarchy of comparative efficacy at post-treatment, efficacy with regard to PTSD symptoms at follow-up, as well as in terms of acceptability (all-cause discontinuation), improvement of anxiety symptoms, improvement of depressive symptoms and quality of life and functional improvement. We expect that the findings could assist patients, clinicians and healthcare providers to make a better informed choice in treatment selection.

There are some limitations in this protocol. First, due to the fact that PTSD commonly co-occurs with other psychiatric comorbidities, we will not exclude trials in which patients with comorbidity were enrolled. This will enhance the generalisability of this study; however, it will also raise the risk of bias for outcomes. Further individual patient data meta-analysis will be helpful to investigate the influence of this factor. Second, some trials may tend to report their favourite outcomes. Although we have predefined the data selection criteria for each outcome, this selection bias could not be completely eliminated.

**Author affiliations**
[1]Department of Neurology, The First Affiliated Hospital of Chongqing Medical University, Chongqing, China
[2]Institute of Neuroscience and the Collaborative Innovation Center for Brain Science, Chongqing Medical University, Chongqing, China
[3]Department of Psychiatry, The First Affiliated Hospital of Chongqing Medical University, Chongqing, China
[4]Department of Psychological Medicine, University of Auckland, Auckland, New Zealand
[5]The Centre of Youth Mental Health, University of Melbourne, Melbourne, Victoria, Australia
[6]Department of Psychology, Harvard University, Cambridge, Massachusetts, USA
[7]Department of Clinical, Neuro and Developmental Psychology, Amsterdam Public Health research institute, Vrije Universiteit Amsterdam, Amsterdam, The Netherlands
[8]Institute for Complementary and Integrative Medicine, University Hospital and University of Zurich, Zurich, Swaziland
[9]Institute of Primary Health Care (BIHAM), University of Bern, Bern, Swaziland
[10]Department of Child and Adolescent Psychiatry, Hôpital Pitié–Salpétrière, Institut des Systèmes Intelligents et Robotiques, Université Pierre et Marie Curie, Paris, France
[11]Mental Health, Westmead, Western Sydney Local Health District, Parramatta, Australia

**Contributors** PX, YZ and XZ conceived the study and drafted the manuscript. SEH, JRW, PC, JB, CDG, DC and DG assisted in protocol design and revision. YZ, XZ, LY, SY, XJ and TT participated in the search strategy development, and will carry out the most work of study selection, risk of bias assessment and data collection. PX and SEH will help to resolve the disagreements and check the data. CDG, YZ and XZ participated in the design of data synthesis and analysis and will conduct the statistical analyses. All the authors have approved the publication of the protocol.

**Funding** This study was funded by National Key Research and Development Programm of China (Grant No.2017YFA0505700).

**Disclaimer** The funders had no role in the protocol design; the writing of the protocol; or the decision to submit the protocol for publication.

**Competing interests** SEH is an editor of the Cochrane Common Mental Disorders Group and an author on the Cochrane systematic review on treatments for PTSD in young people. During the last 2 years, DC reported past consultation for or the receipt of honoraria from Otsuka, Shire, Lundbeck and Roche. DG is the primary author of two reviews of psychological therapies for children and adolescents who were diagnosed with PTSD or exposed to trauma. YZ, XZ, LY, JRW, PC, JB, CDG, SY, XJ, TT , and PX declare nocompeting interests.

**Patient consent** Not required.

**Provenance and peer review** Not commissioned; externally peer reviewed.

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
