## [Reviewer comments · BMJ Open]

ARTICLE DETAILS

TITLE (PROVISIONAL)	Comparative Efficacy and Acceptability of Psychotherapies for Post-traumatic Stress Disorder in Children and Adolescents: Study Protocol for a Systematic Review and Network Meta-Analysis
AUTHORS	Zhang, Yuqing; Zhou, Xinyu; Yang, Lining; Hetrick, Sarah; Weisz, John; Cuijpers, Pim; Barth, Juergen; Del Giovane, Cinzia; Yuan, Shuai; Cohen, David; Gillies, Donna; Jiang, Xiaofeng; Teng, Teng; Xie, Peng

VERSION 1 – REVIEW

REVIEWER	Roger Mulder Department of Psychological Medicine University of Otago, Christchurch New Zealand
REVIEW RETURNED	07-Nov-2017

GENERAL COMMENTS	The study protocol addresses an important area of child psychiatry namely, the efficacy and acceptability of psychotherapies for post-traumatic stress disorder (PTSD) in children and adolescents. Unfortunately children frequently suffer traumatic events and require treatment. The use of medication is poorly established and difficult to research in children. Therefore psychotherapies would appear to be the primary method of treating PTSD. The authors are a very experienced group with expertise in conducting systematic reviews. The authors note the limited numbers of RCTs directly comparing different types of psychotherapies and in my view justify using a network meta-analyses. The protocol presented is clearly outlined and in particular it is useful to have published primary outcome, secondary outcomes and Table 2 which gives a hierarchy of how PTSD symptom severity and measurement scales will be judged. My only comment would be on why trials in which the number of sessions is less than four would be excluded? My reason for this is that there is the possibility that interventions for PTSD could cause harm. The best evidence for this is single session debriefing. It is also possible that brief interventions might not be helpful in children. Therefore I think the authors need to justify why they have excluded all brief trials even if it is to look for potential harm to participants.
---

REVIEWER	Matej Stuhlec, Ph.D., Pharm.D.
-----------------	--------------------------------

	1. Faculty of Pharmacy Ljubljana Slovenia 2. Ormoz Psychiatric Hospital, Slovenia
REVIEW RETURNED	18-Nov-2017

GENERAL COMMENTS	The authors have written about an interesting topic: »Comparative Efficacy and Acceptability of Psychotherapies for Post-traumatic Stress Disorder in Children and Adolescents: Study Protocol for a Systematic Review and Network Meta-Analysis«. This is an important paper, however there are some data on this important topic, although this is a first network meta-analysis within this field. The authors make a strong case for the need for a network meta-analysis and have proposed to use many state of the science procedures for searching the literature and analyzing the resulting data. This meta-analysis is an upgrade of the previous meta-analysis, especially Cochrane Database Syst Rev. 2016 Oct 11;10:CD012371. The paper provides important information from evidence based medicine that could be considered a bridge between real practice (treatment) and guidelines (recommendations), although there are some very important limitations, which should be addressed. The purpose of the research is well defined. Generally the paper has medium to high issues with the standard of writing. However, in the current form presented, it requires a revision before consideration for publication. The manuscript could be strengthened by attending to the following matters: TITLE: Comparative Efficacy and Acceptability of Psychotherapies for Post-traumatic Stress Disorder in Children and Adolescents: Study Protocol for a Systematic Review and Network Meta-Analysis According to the MEDLINE searching this title is appropriate. Abstract The authors should specify which diagnostic criteria were included in this research. This can have an important impact on the final results (e.g. effect sizes). Were study duration limited? Please specify. BACKGROUND Major comments The authors wrote many different options about PTSD, although medications (e.g. pharmacotherapy) have not been described very in detail. According to the some trials and reviews, some medications have very good results in PTSD treatment. Fluoxetine, paroxetine and venlafaxine may be considered as potential treatments for the disorder. Please use the following paper: »Pharmacotherapy for post-traumatic stress disorder: systematic review and meta-analysis. Br J Psychiatry. 2015 Feb;206(2):93-100. doi: 10.1192/bjp.bp.114.148551« (lines 34-50, page 8/30). The authors wrote. »However, due to the limited number of randomised controlled trials (RCTs) directly comparing different types of psychotherapies, conventional meta-analyses cannot provide a clear answer regarding the best choice for initial treatment, nor with regard to a hierarchy of these psychotherapies.« This is not completely true, because RCTs are also necessary for network meta-analysis. I suggest that the authors modify this sentence. If few RCTs are available it means that results have also wider confidence interval, which means that results should be interpret with a great caution. According to the available data one of the aim of this research could be also the following: According to the lack of data where efficacy of these interventions have been proven, the aim of this research was to add an additional data on psychotherapy efficacy in this important population next to the previous meta-analysis. In this point of view
---

the authors should comment what is new in their paper according to the papers given above.

Usually the guidelines are written according to the evidence based medicine (EBM), where the meta-analyses have the very important role and with these meta-analyses we are getting closer to the evidence level, which is seen only for few medications. I think this is very important aim of this research (clinical implication).

Minor comment

The authors should connect the first and second sentences, because this (Trauma is common in children and adolescents) is well known and it is not sound very scientific (Page 7/30, lines 6-7). Instead of a word »common disease« the authors should use well known epidemiological data (e.g. worldwide prevalence).

I think that the authors should state that PTSD can have an impact on QALYs, which is more important than subjective word »well-being« (lines 31-33). The results of well designed meta-analyses could be used for pharmacoeconomics and calculations (e.g. number of wasted treatments).

The authors should use a word major depressive disorders instead of depression. In all trials MDD has been used and therefore is more appropriate. Depression could be also a symptom (residual) of some other disorders. Please modify in the text (MDD) (lines 26-27).

METHODS

Criteria for included studies

Major comments

Why cluster RCTs were included? This can be a source of bias, especially if patients were randomized from different locations (e.g. patients from the same hospital are more likely to be similar than 2 patients from the 2 different hospitals in terms of outcomes!).

How the authors minimized e.g. "learning" effect in »cross-over trials«, which is especially problematic with psychotherapy. Please make comments on these important topic.

Studies where it was not clear what happened to the patients who withdrew from the study were also excluded? Please clarify this important issue and add an additional text to the manuscript.

Was PICOS method used in this research?

Types of participants

Major comments

Why authors included patients with comorbidity? This is a huge limitation, which should be deeply discussed. In this case I suggest that the authors provide a table with detail % of patients included in each study. However, I would suggest to remove these trials. For example in medication treatment venlafaxine is more efficacious than escitalopram within patients with MDD and ADHD. About 30-50 % patients with ADHD have also MDD or anxiety disorders and therefore this conclusion can have a great impact on the final drug efficacy. Psychotherapy is also not equally effective for MDD, ADHD and PTSD.

Types of interventions

Major comments

Were the authors excluded RCTs where medications were used in the past (pre-study period)?

Were the authors included only treatment naive patients or they mixed RCTs? This can have an impact on the final results.

What was the limitation (duration) time of psychotherapy in RCTs?

Control group is especially problematic in RCTs where psychotherapy has been included. How the authors checked differences in placebos among different RCTs? Some placebo

	groups can have already »psychotherapeutical conditions«. This can have an impact on the final results. In view of the efforts to disseminate evidence-based techniques, I suggest the authors reconsider how they define treatment as usual (TAU) and how they might use it as a control condition. How the authors exclude different psychotherapy standards in different countries? This is very important to avoid serious bias. Types of outcome measures Major comments Usually we have the brief data from scales' differences between finish and start of the trials. However in some trials we have only number of participants to finish these trials successfully according to the defined outcomes (e.g. in 50% reduction in HAM-D17). How the authors will convert different outcomes to appropriate numbers? Please make comments on this question. Different outcomes can be a source of pharmaceutical marketing bias (e.g. favourite research for some companies). With careful consideration before research the authors can reduce these type of bias. If the authors will include the 3 different results from the same trials, 'trials bias' should be calculated and discussed in the discussion (e.g. placebo patients are the same in all 3 different results). Please make comments on this question and discuss in the limitations part of the discussion section. Search strategy Minor comment Were the presentations at EPA, ECNP, APA and ESCAP congresses included? REFERENCES Please check the references again. There are some mistakes (e.g. - or --).
--	---

VERSION 1 – AUTHOR RESPONSE

Editorial Requests:

- We previously noted that the PROSPERO database indicated that this study was already complete. However, you replied as follows: "We did register this project on November 19, 2016 and plan to complete this project in April 2017. However, due to the delay of our other research and a long time discussion of this protocol within the international co-authors, we have confirmed the final protocol until now. Thus, we have not started to collect data. We expect to finish collecting data on April 30, 2018. We have updated the record of this project in PROSPERO database yesterday, however the updated information has not been shown until now. If you have any other questions, please contact me anytime."

The PROSPERO database still indicates that this study was complete on 1st April 2017. Please see: https://www.crd.york.ac.uk/PROSPERO/display_record.php?RecordID=51786 Therefore, can you please check/ update the registry details?

Authors' response

Thanks for the reminding. Our update request has been assessed until November 23, 2017. Now the updated content can be seen. Please see: https://www.crd.york.ac.uk/PROSPERO/display_record.php?RecordID=51786

- Please include the full search strategy as a supplementary file, and refer to this in the methods section, rather than including the full search strategy in the main body of the manuscript.

Authors' response

Thanks. This has been done in the revised manuscript.

Reviewers' Comments to Author:

Reviewer: 1

Reviewer Name: Roger Mulder

Institution and Country: Department of Psychological Medicine, University of Otago, Christchurch, New Zealand

Competing Interests: None declared.

The study protocol addresses an important area of child psychiatry namely, the efficacy and acceptability of psychotherapies for post-traumatic stress disorder (PTSD) in children and adolescents.

Unfortunately children frequently suffer traumatic events and require treatment. The use of medication is poorly established and difficult to research in children. Therefore psychotherapies would appear to be the primary method of treating PTSD.

The authors are a very experienced group with expertise in conducting systematic reviews.

The authors note the limited numbers of RCTs directly comparing different types of psychotherapies and in my view justify using a network meta-analysis. The protocol presented is clearly outlined and in particular it is useful to have published primary outcome, secondary outcomes and Table 2 which gives a hierarchy of how PTSD symptom severity and measurement scales will be judged.

Authors' response

Thanks for the positive comments.

My only comment would be on why trials in which the number of sessions is less than four would be excluded? My reason for this is that there is the possibility that interventions for PTSD could cause harm. The best evidence for this is single session debriefing. It is also possible that brief interventions might not be helpful in children. Therefore I think the authors need to justify why they have excluded all brief trials even if it is to look for potential harm to participants.

Authors' response

Thanks. We previously limited the number of sessions because we want to reduce heterogeneity between included trials. However, after re-consider this point and discuss among the research team, we are agree with the reviewer and decide to not limit number of sessions. We have added the following words: "Study duration and the number of treatment sessions will not be limited." (Types of studies section, page 8)

Reviewer: 2

Reviewer Name: Matej Stuhec, Ph.D., Pharm.D.

Institution and Country: 1. Faculty of Pharmacy Ljubljana Slovenia; 2. Ormoz Psychiatric Hospital, Slovenia

Competing Interests: None declared

The authors have written about an interesting topic: »Comparative Efficacy and Acceptability of Psychotherapies for Post-traumatic Stress Disorder in Children and Adolescents: Study Protocol for a Systematic Review and Network Meta-Analysis«. This is an important paper, however there are some

data on this important topic, although this is a first network meta-analysis within this field. The authors make a strong case for the need for a network meta-analysis and have proposed to use many state of the science procedures for searching the literature and analyzing the resulting data. This meta-analysis is an upgrade of the previous meta-analysis, especially Cochrane Database Syst Rev. 2016 Oct 11;10:CD012371. The paper provides important information from evidence based medicine that could be considered a bridge between real practice (treatment) and guidelines (recommendations), although there are some very important limitations, which should be addressed. The purpose of the research is well defined. Generally the paper has medium to high issues with the standard of writing. However, in the current form presented, it requires a revision before consideration for publication. The manuscript could be strengthened by attending to the following matters:
TITLE: Comparative Efficacy and Acceptability of Psychotherapies for Post-traumatic Stress Disorder in Children and Adolescents: Study Protocol for a Systematic Review and Network Meta-Analysis
According to the MEDLINE searching this title is appropriate.

Authors' response

Thanks. (No comments are needed)

Abstract

The authors should specify which diagnostic criteria were included in this research. This can have an important impact on the final results (e.g. effect sizes).

Authors' response

Yes. We have added the following words in the revised manuscript: "...children and adolescents (18 years old or less) diagnosed with full or subclinical PTSD will be included". (Abstract section, page 3)

Were study duration limited? Please specify.

Authors' response

We will not limit study duration. In addition, as the suggestion by reviewer 1, we will not limit treatment sessions. We have added the following words to abstract: "Study duration and the number of treatment sessions will not be limited." (Abstract section, page 3)

BACKGROUND

Major comments

The authors wrote many different options about PTSD, although medications (e.g. pharmacotherapy) have not been described very in detail. According to the some trials and reviews, some medications have very good results in PTSD treatment. Fluoxetine, paroxetine and venlafaxine may be considered as potential treatments for the disorder. Please use the following paper: »Pharmacotherapy for post-traumatic stress disorder: systematic review and meta-analysis. Br J Psychiatry. 2015 Feb;206(2):93-100. doi: 10.1192/bjp.bp.114.148551 « (lines 34-50, page 8/30).

Authors' response

Thanks, however this paper did not include children and adolescents with PTSD. We have added the following words: "Although some medications (e.g., fluoxetine, paroxetine and venlafaxine) have shown a significant but small superiority over placebo (mean effect size=0.23) in PTSD treatment for adults, their use in children and adolescents is still limited because..." (Background section, page 6)

The authors wrote. »However, due to the limited number of randomised controlled trials (RCTs) directly comparing different types of psychotherapies, conventional meta-analyses cannot provide a clear answer regarding the best choice for initial treatment, nor with regard to a hierarchy of these psychotherapies.« This is not completely true, because RCTs are also necessary for network meta-

analysis. I suggest that the authors modify this sentence. If few RCTs are available it means that results have also wider confidence interval, which means that results should be interpreted with a great caution.

Authors' response

Yes, I agree. We have revised this sentence as "However, due to methodological limitations, conventional meta-analyses cannot simultaneously compare all these treatments. Therefore, they cannot provide a clear answer regarding the best choice for initial treatment, nor can they provide a hierarchy of these psychotherapies." (Background section, page 7)

According to the available data one of the aim of this research could be also the following: According to the lack of data where efficacy of these interventions have been proven, the aim of this research was to add an additional data on psychotherapy efficacy in this important population next to the previous meta-analysis. In this point of view the authors should comment what is new in their paper according to the papers given above.

Authors' response

Thanks. In this paper, we will add updated data, and perform developed statistical methodology. We have added the following comments in the revised manuscript: "In the present study we describe the methods to undertake a network meta-analysis to complement those previous reports that will focus on the efficacy of psychotherapies for children and adolescents with PTSD." (Background section, page 8)

Usually the guidelines are written according to the evidence based medicine (EBM), where the meta-analyses have the very important role and with these meta-analyses we are getting closer to la evidence level, which is seen only for few medications. I think this is very important aim of this research (clinical implication).

Authors' response

Thanks. We have added the following words: "By performing a well-designed Bayesian network meta-analysis, we aim to provide a higher level of evidence that can inform clinical guidelines for PTSD treatment in young people." (Background section, page 8)

Minor comment

The authors should connect the first and second sentences, because this (Trauma is common in children and adolescents) is well known and it is not sound very scientific (Page 7/30, lines 6-7). Instead of a word »common disease« the authors should use well known epidemiological data (e.g. worldwide prevalence).

Authors' response

Thanks. We have connected and modified the first and second sentences as follows: "Many children and adolescents are exposed to trauma, with more than two thirds reporting at least 1 traumatic event by 16 years of age. Post-traumatic stress disorder (PTSD) is one of the most common mental disorders among children and adolescents who have experienced trauma, with an overall prevalence of 15.9%." (Background section, page 6). Due to the fact that trauma exposure is an essential factor in order to be able to diagnose PTSD, most studies in this field use the prevalence of PTSD in people who have exposed to trauma. The epidemiological data we presented is reported by a meta-analysis pooling 72 peer-reviewed articles on 43 independent samples worldwide. This is a relative new and reliable data, therefore we prefer to still use this data.

I think that the authors should state that PTSD can have an impact on QALYs, which is more important than subjective word »well-being« (lines 31-33). The results of well designed meta-analyses could be used for pharmacoeconomics and calculations (e.g. number of wasted treatments).

Authors' response

We have added the following words in the revised manuscript: "PTSD diagnosis is accompanied by a significant reduction in quality of life, and it is estimated that successful treatment could save 2.05 quality adjusted life years (QALYs) per child or adolescent with PTSD". (Background section, page 6)

The authors should use a word major depressive disorders instead of depression. In all trials MDD has been used and therefore is more appropriate. Depression could be also a symptom (residual) of some other disorders. Please modify in the text (MDD) (lines 26-27).

Authors' response

Thanks. This has been modified in the revised manuscript.

METHODS

Criteria for included studies

Major comments

Why cluster RCTs were included? This can be a source of bias, especially if patients were randomized from different locations (e.g. patients from the same hospital are more likely to be similar than 2 patients from the 2 different hospitals in terms of outcomes!).

Authors' response

Although cluster RCTs would be a source of bias, we would not arbitrarily exclude all these studies, and will use the new tool for risk of bias assessment (Cochrane Collaboration's Risk of bias 2.0) to assess the risk of bias from this factor. The new tool which will be used in this study includes an additional domain for cluster-randomised trials ((1b) Bias arising from the timing of identification and recruitment of individual participants). According to this tool, we could be easier to accurately assess the risk of bias of cluster RCTs and then adjust the level of evidence appropriately.

How the authors minimized e.g. "learning" effect in »cross-over trials«, which is especially problematic with psychotherapy. Please make comments on these important topic.

Authors' response

We have stated that "only the results from the first randomisation period will be considered in the cross-over trials". Therefore the "learning" effect could be avoided.

Studies where it was not clear what happened to the patients who withdrew from the study were also excluded? Please clarify this important issue and add an additional text to the manuscript.

Authors' response

These studies will not be excluded. We have added the following text in the revised manuscript: "Studies where it is not clear what happened to the patients who withdrew from the study will not be excluded, and all these patients will be counted as all-cause discontinuation". (Types of participants section, page 9)

Was PICOS method used in this research?

Authors' response

Yes, PICOS method was described in section of "Types of studies", "Types of participants", "Types of interventions", and "Types of outcome measures" in the manuscript.

Types of participants

Major comments

Why authors included patients with comorbidity? This is a huge limitation, which should be deeply discussed. In this case I suggest that the authors provide a table with detail % of patients included in each study. However, I would suggest to remove these trials. For example in medication treatment venlafaxine is more efficacious than escitalopram within patients with MDD and ADHD. About 30-50 % patients with ADHD have also MDD or anxiety disorders and therefore this conclusion can have a great impact on the final drug efficacy. Psychotherapy is also not equally effective for MDD, ADHD and PTSD.

Authors' response

Yes, patients with comorbidity could be a source of bias. However, comorbidity is very common in children and adolescents with PTSD. If we exclude patients with comorbidity, the generalizability of our study will be largely limited. Moreover, according to our preliminary data, many studies did not report comorbidity. We cannot know whether they did not include patients with comorbidity, or they just had not assessed comorbidity. Thus, we prefer not to exclude patients with comorbidity. And we think further individual patient data meta-analysis will be helpful to address this issue. We have added this as a limitation of this study: "There are some limitations in this protocol. First, due to the fact that PTSD commonly co-occurs with other psychiatric comorbidities, we will not exclude trials in which patients with comorbidity were enrolled. This will enhance the generalizability of this study; however, it will also raise the risk of bias for outcomes". (Discussion section, page 15)

Types of interventions

Major comments

Were the authors excluded RCTs where medications were used in the past (pre-study period)?

Authors' response

We have added the following words: "We will not exclude studies that enrolled patients who had used medications in the past, provided that their medication status was not changed for at least one month prior to study entry and for the study period". (Types of interventions section, page 10)

Were the authors included only treatment naive patients or they mixed RCTs? This can an impact on the final results.

Authors' response

As we responded above, We will not exclude studies that enrolled patients who had used medications in the past. Therefore, studies will not be required to include only treatment naive patients".

What was the limitation (duration) time in RCTs?

Authors' response

As we responded above (#3 response), Study duration and the number of treatment sessions will not be limited." (Types of studies section, page 8)

Control group is especially problematic in RCTs where psychotherapy has been included. How the authors checked differences in placebos among different RCTs? Some placebo groups can have already »psychotherapeutical conditions«. This can have an impact on the final results.

Authors' response

The control conditions we predefined have their major principles, which have been described in table 1. According to the principles described in this table, two reviewers will independently perform the

classification of all conditions in each trial. Any disagreements will be discussed among the review team, and any unclear information will be requested from the relevant authors. This has been added in the revised manuscript. (Types of interventions section, page 10).

It is indeed that some placebo groups can have already “psychotherapeutical conditions”. After re-discussing among the research team, we think that it is more appropriate to classify supportive therapy (ST) into intervention condition because most studies view this condition as a non-specific control intervention. It is noted that this classification will not affect the final network estimates, because all nodes in network meta-analysis will be compared simultaneously, despite whether they are interventions or control conditions.

In view of the efforts to disseminate evidence-based techniques, I suggest the authors reconsider how they define treatment as usual (TAU) and how they might use it as a control condition.

Authors' response

Treatment as usual (TAU) is not a structured condition. The components of TAU is varied in different trials. Therefore the definition could not be very specific. We have added some descriptions to the definition as follows: “TAU is often described as "usual care" or "usual community treatment" in trials, which may include any components of psychotherapy or pharmacotherapy for PTSD. It is not considered to be structured intervention but may have some treatment effects”. (Table 1, page 24) Based on our preliminary data, TAU is usually used as a control condition to compare with an active psychotherapy in RCTs. In addition, as we responded above, all nodes in network meta-analysis, despite of interventions or control conditions, will be compared simultaneously. Thus, whether we use TAU as an intervention or a control condition, the network estimates will not be influenced.

How the authors exclude different psychotherapy standards in different countries? This is very important to avoid serious bias.

Authors' response

As we responded above, two reviewers will independently perform the classification of all conditions in each trial according to the predefined principles described in table 1. Any disagreements will be discussed among the review team, and any unclear information will be requested from the relevant authors. (Types of interventions section, page 10). Through these procedures, we hope to exclude the largest possible influence of different psychotherapy standards in different countries.

Types of outcome measures

Major comments

Usually we have the brief data from scales' differences between finish and start of the trials. However in some trials we have only number of participants to finish these trials successfully according to the defined outcomes (e.g. in 50% reduction in HAM-D17). How the authors will convert different outcomes to appropriate numbers? Please make comments on this question.

Authors' response

We have added the following statement in the revised manuscript: “Where dichotomous efficacy outcomes, instead of continuous scores, are reported in a trial, we will contact the relevant authors to request the data we need. If they don't respond the data will not be used”. (Types of outcome measures section, page 10)

Different outcomes can be a source of pharmaceutical marketing bias (e.g. favourite research for some companies). With careful consideration before research the authors can reduce these type of bias. If the authors will include the 3 different results from the same trials, 'trials bias' should be calculated and discussed in the discussion (e.g. placebo patients are the same in all 3 different

results). Please make comments on this question and discuss in the limitations part of the discussion section.

Authors' response

Pharmaceutical companies are rarely involved in psychotherapy trials. However, it is possible that some researchers selectively reported their favourite outcomes. Thus, in order to reduce this selection bias, we have predefined the data selection criteria (types of data selection, measurement selection) of each outcome. Therefore, we will not include the 3 different results from the same trials. We have added this to the limitations part of the discussion section as follows: "Second, some trials may tend to report their favourite outcomes. Although we have predefined the data selection criteria for each outcome, this selection bias could not be completely eliminated". (Discussion section, page 15).

Search strategy

Minor comment

Were the presentations at EPA, ECNP, APA and ESCAP congresses included?

Authors' response

No, these presentations were not included in our previous search strategy. We have now added them in the revised manuscript. (Search strategy, page 12)

REFERENCES

Please check the references again. There are some mistakes (e.g.- or --).

Authors' response

We are sorry for this. These mistakes have now been addressed.

VERSION 2 – REVIEW

REVIEWER	Matej Stuhec, Ph.D., Pharm.D. Department of Biopharmacy and Pharmacokinetics, School of Pharmacy, University of Ljubljana, Ljubljana, Slovenia. Department for Clinical Pharmacy, Psychiatric Hospital Ormoz, Ptujška Cesta 33, Ormoz, Slovenia.
REVIEW RETURNED	31-Dec-2017

GENERAL COMMENTS	The authors accepted all my recommendations. This version of manuscript has been improved. In my point of view it is appropriate for publication, when some remarks/recommendations would be addressed. 1. Introduction. General Prevalence (2nd sentence). Please specify prevalence more in detail (e.g. obtained from meta-analyses, trials, epidemiological data, registry data etc.). This can have an important impact on the final prevalence result (please check details for ADHD children and adolescents epidemiology: Croat Med J. 2015 Apr;56(2):159-65.). This can overrate(under) a global burden of disease. ADHD = please explain abbreviation when used first time I suggest that the authors avoid a word: young people because usually RCTs have been done with children and adolescents. Young people can mean only children or with adolescents. Please modify. 2. Methods.
--

	Types of outcome measures Primary outcome If they don't respond the data will not be used. Didn't respond? Secondary outcomes »end point score on anxiety symptom severity rating scales« »4. Depressive symptoms, as measured by the end point score on depressive symptom severity rating scales.« This could be a source of bias, because if the authors used only »end point« there are no data about their baseline status (e.g. baseline points). More appropriate would be »differences in points (end-baseline)« (e.g. if we have more depressed patients we profit more from the treatment strategy than »medium« depressed patients. It has been already discussed in many previous publications). Please comment.
--	--

REVIEWER	Roger Mulder University of Otago Christchurch New Zealand
REVIEW RETURNED	03-Jan-2018

GENERAL COMMENTS	The authors have answered my major concern and I think that this increases the clinical relevance of their work. I think that the possibility that treatments-including psychotherapies- may do harm needs to be kept in mind
---

VERSION 2 – AUTHOR RESPONSE

Editorial Request:

Regarding the supplementary information: can you please amend the title - "The full search strategy for database"- to clarify which database you are referring to?

Authors' response

Thanks. We have amended the title as "the full search strategy for PubMed", and revised the corresponding words in the manuscript.

Reviewers' Comments to Author:

Reviewer: 1

Reviewer Name: Roger Mulder

Institution and Country: University of Otago, Christchurch, New Zealand

Competing Interests: none

The authors have answered my major concern and I think that this increases the clinical relevance of their work. I think that the possibility that treatments-including psychotherapies- may do harm needs to be kept in mind.

Authors' response

Thanks very much. We appreciate #1 reviewer's kind suggestion.

Reviewer: 2

Reviewer Name: Matej Stuhec, Ph.D., Pharm.D.

Institution and Country: Department of Biopharmacy and Pharmacokinetics, School of Pharmacy, University of Ljubljana, Ljubljana, Slovenia. Department for Clinical Pharmacy, Psychiatric Hospital Ormoz, Ptujška Cesta 33, Ormoz, Slovenia.

Competing Interests: None declared

The authors accepted all my recommendations. This version of manuscript has been improved. In my point of view it is appropriate for publication, when some remarks/recommendations would be addressed.

1. Introduction.

General Prevalence (2nd sentence). Please specify prevalence more in detail (e.g. obtained from meta-analyses, trials, epidemiological data, registry data etc.). This can have an important impact on the final prevalence result (please check details for ADHD children and adolescents epidemiology: Croat Med J. 2015 Apr;56(2):159-65.). This can overrate(under) a global burden of disease.

Authors' response

Yes. We have amended this sentence as "Data from a meta-analysis pooling 72 articles with 3563 trauma-exposed children and adolescents show that the overall rate of PTSD was 15.9%". (page 6)

ADHD = please explain abbreviation when used first time.

Authors' response

Thanks. The full name of this abbreviation has been added. (page 6)

I suggest that the authors avoid a word: young people because usually RCTs have been done with children and adolescents. Young people can mean only children or with adolescents. Please modify.

Authors' response

Thanks. We have amended all these words.

2. Methods.

Types of outcome measures

Primary outcome

If they don't respond the data will not be used. Didn't respond?

Authors' response

Thanks. We have corrected this.

Secondary outcomes

»end point score on anxiety symptom severity rating scales«

»4. Depressive symptoms, as measured by the end point score on depressive symptom severity rating scales.«

This could be a source of bias, because if the authors used only »end point« there are no data about their baseline status (e.g. baseline points). More appropriate would be »differences in points (end-baseline)« (e.g. if we have more depressed patients we profit more from the treatment strategy than »medium« depressed patients. It has been already discussed in many previous publications). Please comment.

Authors' response

Yes. However, change score is also a source of bias because the scores on baseline and post-test are not independent of each other, and the value for the correlation should be used in the calculation

of the SMD, while this value is typically not known. This problem can lead to considerable errors in the estimation of the SMDs. More details please see: *Epidemiol Psychiatr Sci.* 2017 Aug;26(4):364-368. This issue has been discussed many times among our research group, and we finally decide to choose end point score. We will carefully check whether the severity of PTSD, anxiety, and depressive symptom is balanced at baseline in each study. If not, we will rank it as high risk of bias in the term "other bias" in Cochrane Collaboration's Risk of bias tool.

VERSION 3 – REVIEW

REVIEWER	Matej Stuhec, Ph.D., Pharm.D. Department of Biopharmacy and Pharmacokinetics, School of Pharmacy, University of Ljubljana, Ljubljana, Slovenia. Department for Clinical Pharmacy, Psychiatric Hospital Ormoz, Ptujška Cesta 33, Ormoz, Slovenia
REVIEW RETURNED	26-Jan-2018
GENERAL COMMENTS	The authors accepted all my remarks/recommendations and therefore I suggest to accept this paper.